

# Impacts of land-use change and elevated CO₂ on the interannual variations and seasonal cycles of gross primary productivity in China

Binghao Jia[1,2,3], Xin Luo[1], Ximing Cai[3], Atul Jain[4], Deborah N Huntzinger[5], Zhenghui Xie[1], Ning Zeng[1,6], Jiafu Mao[7], Xiaoying Shi[7], Akihiko Ito[8], Yaxing Wei[7], Hanqin Tian[9], Benjamin Poulter[10], Dan Hayes[11], Kevin Schaefer[12]

[1]State Key Laboratory of Numerical Modeling for Atmospheric Sciences and Geophysical Fluid Dynamics (LASG), Institute of Atmospheric Physics, Chinese Academy of Sciences, Beijing, China
[2]Key Lab of Guangdong for Utilization of Remote Sensing and Geographical Information System,Guangzhou Institute of Geography, Guangzhou, China
[3]Ven Te Chow Hydrosystems Laboratory, Department of Civil and Environmental Engineering, University of Illinois at Urbana-Champaign, Urbana, Illinois, USA
[4]Department of Atmospheric Sciences, University of Illinois at Urbana-Champaign, Urbana, Illinois, USA
[5]School of Earth Sciences and Environmental Sustainability and Department of Civil Engineering, Construction Management, and Environmental Engineering, Northern Arizona University, Flagstaff, Arizona, USA
[6]Department of Atmospheric and Oceanic Science, University of Maryland, College Park, Maryland, USA
[7]Environmental Sciences Division, Climate Change Science Institute, Oak Ridge National Laboratory, Oak Ridge, Tennessee, USA
[8]Center for Global Environmental Research, National Institute for Environmental Studies, Tsukuba, Japan
[9]International Center for Climate and Global Change Research and School of Forestry and Wildlife Sciences, Auburn University, Auburn, Alabama, USA
[10]NASA GSFC, Biospheric Sciences Laboratory, Greenbelt, MD, USA
[11]School of Forest Resources, University of Maine, Orno, Maine, USA
[12]National Snow and Ice Data Center, Cooperative Institute for Research in Environmental Sciences, University of Colorado at Boulder, Boulder, Colorado, USA

*Correspondence to*: Binghao Jia (bhjia@mail.iap.ac.cn)

**Abstract.** Climate change, rising $CO_2$ concentration, and land use and land cover change (LULCC) are primary driving forces for terrestrial gross primary productivity (GPP), but their impacts on the temporal changes in GPP are confounded. In this study, the effects of the three main factors on the interannual variation (IAV) and seasonal cycle amplitude (SCA) of GPP in China were investigated using 12 terrestrial biosphere models from the Multi-scale Synthesis and Terrestrial Model Intercomparison Project. The simulated ensemble mean value of China's GPP, driven by common climate forcing, LULCC, and $CO_2$ data, was found to be 7.4±1.8 Pg C yr$^{-1}$, which was in close agreement with the independent upscaling GPP estimate (7.1 Pg C yr$^{-1}$). In general, climate was the dominant control factor of the annual trends, IAV, and seasonality of China's GPP. The overall rising



$CO_2$ led to enhanced plant photosynthesis, thus increasing annual mean and IAV of China's total GPP, especially in northeastern and southern China where vegetation is dense. LULCC decreased the IAV of China's total GPP by ~7%, whereas rising $CO_2$ induced an increase of 8%. Compared to climate change and elevated $CO_2$, LULCC showed less contributions to GPP's temporal variation and its

impact acted locally, mainly in southwestern China. Furthermore, this study also examined subregional contributions to the temporal changes in China's total GPP. Southern and southeastern China showed higher contributions to China's annual GPP, whereas southwestern and central parts of China explained larger fractions of the IAV in China's GPP.

**Keywords:** land-use and land-cover change, MsTMIP, terrestrial biosphere models, gross primary

productivity, interannual variation.

## 1. Introduction

Terrestrial ecosystems can function as a major sink in the global carbon cycle, potentially offsetting a significant amount of anthropogenic carbon emissions (Le Quéré *et al.*, 2017). Gross primary productivity (GPP) is the major driver of terrestrial ecosystem carbon storage and plays a key

role in terrestrial carbon cycle (Yuan *et al.*, 2010; Mao *et al.*, 2012; Piao *et al.*, 2013; Anav *et al.*, 2015; Zhou *et al.*, 2016; Ito *et al.*, 2017). Therefore, understanding the spatial-temporal patterns of terrestrial ecosystem GPP has been a research focus in quantifying the global carbon cycle (Anav *et al.*, 2015; Zhou *et al.*, 2016; Chen *et al.*, 2017). However, GPP is susceptible to $CO_2$ concentration and human interference (primarily land use and land cover change (hereafter LULCC)) besides climate change

(Friedlingstein *et al.*, 2010; Ciais *et al.*, 2013; Li et al., 2015), which complicates the quantification of the impacts.

Atmospheric $CO_2$ concentration has increased by ~40% from 1750 to 2011 (IPCC, 2013). Several studies have examined the effect of rising $CO_2$ concentration on global terrestrial carbon uptake (Piao *et al.*, 2013; Schimel *et al.*, 2014; Ito *et al.*, 2016). Schimel *et al.* (2014) found that about 60% of the

uptake by terrestrial ecosystem was due to raising atmospheric $CO_2$. Simulations from a coupled earth system indicated that $CO_2$ fertilization increased the global net primary productivity ~2.3 Pg C yr$^{-1}$ between 1850 and 2005 (Devaraju *et al.*, 2016). It suggests that the impact of raising $CO_2$ on land carbon sink may be a negative feedback to future climate (Schimel *et al.*, 2014). However, the extent to which $CO_2$ fertilization is responsible for current and future terrestrial carbon storage is still unclear

(Zaehle *et al.*, 2010; IPCC, 2013).

Anthropogenic LULCC also has a large effect on terrestrial carbon cycles, including the "net effect" of $CO_2$ sources (e.g., deforestation, logging, harvesting, and other direct human activities) and





CO$_2$ sinks (e.g., afforestation and vegetation regrowth following land disturbance) (Brovkin *et al.*, 2004; Boysen *et al.*, 2014; Pongratz *et al.*, 2014; Houghton *et al.*, 2017). IPCC (2013) pointed out that LULCC-associated CO$_2$ emissions have contributed ~180 ± 80 Pg C to cumulative anthropogenic CO$_2$ emissions (one third of total anthropogenic CO$_2$ emissions) since 1750. As indicated by Le Quéré *et al.* (2017), CO$_2$ emissions from LULCC at the global scale have remained relatively constant, at around 1.3±0.7 Pg C yr$^{-1}$, over the past half-century. However, regional CO$_2$ emissions showed different characteristics (Houghton *et al.*, 2017).

During the past decades, China has experienced tremendous LULCC as a result of continued population growth and intensified human development against a broad background of climate change (Piao *et al.*, 2009; Liu and Tian, 2010; Xiao *et al.*, 2015; Li et al., 2015; Zhang et al., 2016). These massive LULCCs have made a significant contribution to regional and global carbon sinks during the past few decades (Guo *et al.*, 2013; Fang *et al.*, 2014; Xiao *et al.*, 2015; Li et al., 2015). Hence, studies on the impacts of LULCC on GPP in China have important theoretical and practical value for understanding the temporal-spatial patterns of terrestrial carbon cycle and forecasting their response to future global and regional changes and human activities (Tian et al., 2011a, 2011b).

However, few studies have adequately explored the impacts of climate change, atmospheric CO$_2$ concentration, and LULCC to interannual and seasonal variations of GPP in China. Moreover, the quantitative contributions of these three factors on GPP in China are still unclear, which urgently needs to be addressed. Although continuous improvements have been achieved for the development of terrestrial biosphere models (TBMs) alongside our deepening understanding of terrestrial carbon cycle process, currents TBMs still have large uncertainties in GPP simulation (Piao *et al.*, 2013; Devaraju *et al.*, 2016; Ito *et al.*, 2016). Multi-model ensemble simulation has been an effective method to reduce the uncertainties induced by TBMs (Schwalm *et al.*, 2015; Liu *et al.*, 2016). Therefore, in the present study, twelve progress-based TBMs from the Multi-scale Synthesis and Terrestrial Model Intercomparison Project (MsTMIP) (Huntzinger *et al.*, 2013; Wei *et al.*, 2014a) were used to investigate the effects of climate change, increasing CO$_2$ concentration and LULCC on the interannual variation and seasonal cycle of GPP in China. The goals of this work were to: (1) investigate the interannual and seasonal variations of GPP in China between 1981 and 2010, (2) quantify the individual influences of climate change, CO$_2$ concentration, and LULCC, and (3) examine the relative contributions of major sub-regions to China's total GPP.



## 2. Materials and methods

### 2.1 Model description and experimental design

Twelve TBMs that participated in the MsTMIP were used in this study: CLM4, CLM4VIC, DLEM, GTEC, ISAM, LPJ-wsl, ORCHIDEE-LSCE, SiB3-JPL, SiB3CASA, TEM6, VEGAS2.1, and VISIT (Huntzinger *et al.*, 2013; Wei *et al.*, 2014a, 2014b). These model simulations all followed the same experimental design. Three sensitivity model simulations were used in this study: SG1, driven by time-varying climate data; SG2, considering the effect of LULCC based on SG1; and SG3, similar to SG2, but using time-varying atmospheric $CO_2$ concentration. In this way, these three experiments can be used to assess the relative contributions of climate change, LULCC, and rising $CO_2$ concentration to temporal changes in GPP (Table S1). All the simulated results have a spatial resolution of $0.5° \times 0.5°$ and are available at https://doi.org/10.3334/ORNLDAAC/1225 (Huntzinger et al., 2018). More detailed descriptions of the experimental design and forcing data sets can be found in the supplemental materials and Huntzinger *et al*. (2013) and Wei *et al*. (2014a, 2014b). The simulated monthly GPP from these 12 models was conducted for the period of 1981–2010. The mean values calculated from these models (hereafter 'ENSEMBLE') were also calculated.

### 2.2 Evaluation data

This study used an observation-driven global monthly gridded GPP product derived from FLUXNET measurements by statistical upscaling with a machine-learning algorithm (Jung *et al*., 2009, 2011) (hereafter referred to as MTE). The MTE GPP product was generated by integrating remote-sensing and meteorological data, and land cover information. It has a spatial resolution of $0.5° \times 0.5°$ and is available between 1982 and 2011. The uncertainty of the MTE data is ~46 g C $m^{-2}$ $yr^{-1}$ (5%), which was calculated using the standard deviation of the 25 model tree ensembles (Jung *et al*., 2011).

### 2.3 Analysis method

The land area of China was divided into nine regions (Fig. 1a) through the consideration of their climate characteristics, plant vegetation types, and geopolitical boundaries (Piao *et al*., 2009, 2010). For the whole of China and each sub-region, interannual variations (IAV), seasonal cycle amplitude (SCA), and GPP trends were analyzed and compared across MsTMIP models and MTE data. The IAV of GPP was defined using the standard deviations of each region's detrended annual time-series data. The SCA of GPP was defined as the difference between the largest and smallest values, indicating the maximum range of oscillation between peak and trough within a calendar year (Ito *et al*., 2016).



The nonparametric Mann-Kendall method was used to determine the statistical significance of trends in Chinese and regional GPP (area-weighted), where the Sen median slope (Sen, 1968) was considered as the trend value in this paper. Trend analysis was based on annual values averaged from monthly values. The relative contribution of each sub-region to the IAV and SCA of China's GPP was

also calculated based on the method proposed by Ahlström *et al.* (2015) and Chen *et al.* (2017). Please see the supplemental material for more information.

## 3. Results

### 3.1 Spatial patterns of GPP over China

In general, the spatial distributions of GPP from MsTMIP models (SG3) agreed well with the

MTE (Fig. S1), with spatial correlation coefficients for most models higher than 0.9. The highest GPP values were observed in southeastern (R7) and southern China (R8) due to the wet climate and high solar radiation, and the smallest GPP values were mainly in arid regions of China (e.g., northwestern China) and the Tibetan Plateau due to adverse conditions for plant photosynthetic activities. But the 12 models still have some differences in the spatial variations of GPP. VISIT showed a lower spatial

correlation with the MTE (0.89) due to its higher GPP in southeastern China and lower values in northeastern China. Compared to the MTE, three models (DLEM, TEM6, and VEGAS2.1) produced lower GPP in northeastern China, and VEGAS2.1 produced higher GPP over northwestern China and the western parts of the Tibetan Plateau. The multi-model ensemble mean (ENSEMBLE) showed the highest spatial correlation with the MTE, suggesting that the ensemble mean best captured MTE spatial

variability.

Figure 2 shows the annual mean GPP over China and each sub-region. The twelve models' estimates of total China GPP were found to diverge, ranging from 4.9 (DLEM) to 9.2 (LPJ-wsl) Pg C yr$^{-1}$ (Fig. 1a), with a standard deviation of 1.8 Pg C yr$^{-1}$. The total China GPP from multi-model ensemble mean was 7.4 Pg C yr$^{-1}$, which was slightly higher than the MTE (7.0 Pg C yr$^{-1}$, Fig. 2a).

The regional sum of GPP in southwestern China from the ENSEMBLE (Fig. 2b) was the highest among all nine regions (1.5 Pg C yr$^{-1}$), followed by southeastern China (1.3 Pg C yr$^{-1}$) and southern China (1.0 Pg C yr$^{-1}$). These top three regions together contributed about 50% of China's GPP (Fig. 2c). However, southern China showed the highest GPP estimates per unit area, at > 2000 g C m$^{-2}$ yr$^{-1}$ (Fig. 2d). The relative contributions of each region to total China GPP from the MTE showed results

similar to the MsTMIP. To understand more thoroughly the underlying mechanisms of GPP changes during 1981–2010, the effects of LULCC and atmospheric $CO_2$ concentration on GPP changes were quantified based on the ensemble mean of the 12 MsTMIP models (Table 1). In general, LULCC (SG2,





7.1 Pg C yr$^{-1}$) decreased annual mean GPP by ~0.2 Pg C yr$^{-1}$ (3% of SG1) compared to SG1 (6.9 Pg C yr$^{-1}$). In contrast, elevated atmospheric $CO_2$ increased GPP by ~0.7 Pg C yr$^{-1}$ (10% of SG1), although this response varied among different sub-regions (Table 1a). These results suggested that rising atmospheric $CO_2$ concentration seems to have a greater effect on annual mean GPP over China than LULCC.

### 3.2 Interannual variations and trends

During 1981−2010, the MTE estimates suggested that the IAV of China's GPP was 0.157 Pg C yr$^{-1}$, but the multi-model ensemble mean values of MsTMIP for the three simulations all showed a slight underestimation (Table 1b). Compared to SG1 (0.099 Pg C yr$^{-1}$), LULCC decreased the IAV by ~0.007 Pg C yr$^{-1}$ (7% of SG1), whereas rising $CO_2$ (SG3) led to an increase (~0.008 Pg C yr$^{-1}$, 8% of SG1). The GPP from SG3 with consideration of LULCC and elevated $CO_2$ increased from 7.1 Pg C yr$^{-1}$ in 1981 to 7.6 Pg C yr$^{-1}$ in 2010, with a significant temporal trend of 0.02 Pg C yr$^{-2}$ ($p < 0.05$). The annual mean GPP values from SG3 exhibited significant increasing trends between 1981–2010 over all regions except for Inner Mongolia (R2, Fig. 3c), with the highest rates of increase over the Tibetan Plateau (R6, Fig. 3g) and southeastern China (R7, Fig. 3h), which were both more than 3.0 Tg C yr$^{-2}$ ($p < 0.05$, 1 Tg C = 0.001 Pg C). Compared to SG1 (red line) with prescribed land cover, LULCC (blue line) decreased GPP trends over all regions, which was mainly related to land conversion including forest-to-crop and shrub-to-crop (Tao *et al*., 2013). On the contrary, elevated $CO_2$ concentration significantly increased plant growth and thus led to more strongly increasing GPP trends (SG3, purple line). Compared to the SG3 simulations of MsTMIP, the MTE estimates appeared to show similar interannual variations (Table 1b). Figure S2 shows the spatial patterns of the correlation coefficients between annual GPP from MsTMIP and MTE. It is found that, compared to SG1 and SG2, SG3 captures the interannual variations in GPP of MTE best, with significantly positive correlations over most areas of China, except over the west of Inner Mongolia and parts of central China and northeast China. The highest correlations mainly occur over the middle of Inner Mongolia and northeast of the Tibetan Plateau. In addition, SG3 has the same trends in GPP (significantly increasing) with MTE for R3, R4, R5, R6, R8, and R9 (Figs. 3d, e, f, g, i, j), except some differences in the magnitude. For example, the SG3 is found to show weaker increasing trend (2.0 Tg C yr$^{-2}$) for northern China and larger one for the Tibetan Plateau than the MTE (4.3 Tg C yr$^{-2}$). For R2 (Fig. 3c), SG1 and SG2 show significant decreasing trend while those for SG3 and MTE are not significant. Similar increasing trend can be found for SG3 and MTE over R7 (Fig. 3h) except that the trend of SG3 is significant. Large differences in the trend of GPP can be observed over R1 (Fig. 3b): SG3 shows significant increasing trend while the GPP of MTE is decreasing. However, the mean values and IAV



of GPP over R1 are close between SG3 and MTE (Table 1a). For the whole China (Fig. 3a), the trend in GPP from the MsTMIP is lower than that of the MTE due to large discrepancies between 1999 and 2002. To further validate the trends in GPP from MsTMIP, we compare their spatial distributions with that from MTE (Fig. S3). Compared to SG1 (Fig. S3a), LULCC lead to a decrease in annual mean

GPP (e.g., many areas with stronger negative trend, Fig. S3b). In contrast, rising atmospheric $CO_2$ concentration significantly strengthens the ascending trend in GPP by increasing the rate of photosynthesis (Fig. S3c). Moreover, SG3 capture the trend in GPP of MTE better than SG1 and SG2, with significantly increasing trends over most areas of China and decreasing trends over the east of Inner Mongolia. However, some discrepancies between SG3 and MTE can be observed over northeast

China, east parts of southwestern China and southeastern China.

Figure 4 shows the regional contributions to the IAV of China's GPP for the three MsTMIP simulations (SG1, SG2, and SG3). The ensemble mean GPP of SG3 over southwestern China was found to explain the largest fraction (17%) of the IAV for China's GPP, followed by central China (15%) and northern China (14%). In contrast, the contributions of southern and southeastern China to

the IAV of China's overall GPP were relatively lower (4% and 11%), even with higher contributions to China's annual mean GPP (Fig. 2c). The relative contributions of each sub-region to the IAV of China's GPP from the ENSEMBLE agreed well with the MTE (within one standard deviation), except for a slight overestimation over southwestern China and an underestimation over central China. The contributions from southwestern, central, and northern China were all high for all three MsMTIP

simulations and MTE, except for a few differences in magnitude. Note the significant uncertainties with large standard deviations among the estimated relative contributions of each sub-region from the twelve MsTMIP models, especially in northern and southwestern China. Compared to SG1, SG2 and SG3 showed similar contributions for each sub-region, suggesting that rising atmospheric $CO_2$ and LULCC have little effect on the relative contribution of each region to the IAV of China's GPP.

However, they modulated the magnitude of the IAV and the annual mean values of China and regional GPP (Table 1).

### 3.3 Seasonal variations and regional contributions

Figure 5 shows the seasonal variations in GPP of overall China and each sub-region from the MsTMIP and MTE. In general, the MsTMIP ensemble mean showed seasonal cycles similar to the

MTE data over China and all sub-regions, with strong correlations ($r > 0.97$), except for northern China (Fig. 5e), where large discrepancies in summer (July and August) could be observed. SG1 and SG2 showed almost the same seasonal variations except for a few differences in summer over southeastern China (Fig. 5j), suggesting that LULCC had few effects on seasonal GPP variation in China. In contrast,





elevated $CO_2$ concentrations produced higher GPP during the growing season through enhancing plant growth rate and thus modulated seasonal GPP variations. Table 1c shows that human activities (e.g., LULCC and elevated $CO_2$ concentration) exerted influences on the SCA of GPP. The difference between the SG2 and SG3 was mainly caused by raising atmospheric $CO_2$ concentrations, whereas

LULCC led to a small discrepancy between the SG1 and SG2. For example, compared to SG1, LULCC decreased the SCA by only ~0.3 Pg C yr$^{-1}$ (3% of SG1), whereas elevated $CO_2$ produced an increase of 1.5 Pg C yr$^{-1}$ (14% of SG1). Meanwhile, the SCAs of China's GPP (11.1−12.3 Pg C yr$^{-1}$, Table 1c) from ENSEMBLE were detected with only slight underestimation compared to the MTE data (13.6 Pg C yr$^{-1}$).

Next, the regional contributions to the seasonality of China's GPP were examined for MsTMIP (SG1, SG2, and SG3) and MTE (Fig. 6). The ensemble mean GPP of SG3 (Fig. 6c) over northeastern China explained the largest fraction (20%) of the seasonality of China's GPP, followed by the Tibetan Plateau (16%), and southwestern China (15%). This could be explained because the GPP in these regions had strong seasonal cycles (Figs. 5b, 5g, and 5j). In contrast, the contributions of southern and

southeastern China to the seasonal cycle of China's GPP were relatively low (3% and 8% respectively). The relative regional contributions to the seasonal dynamics of China's GPP from MsTMIP agreed well with MTE (within one standard deviation). The contributions from southwestern, central, and northern China were all high for all three MsTMIP simulations and MTE, except for a few differences in magnitude. Note the significant uncertainties, with large standard deviations, in northern and

northwestern China among the estimated relative regional contributions from the 12 MsTMIP models. Compared to SG1, SG2 and SG3 showed similar contributions for each sub-region, suggesting that atmospheric $CO_2$ and LULCC had little effect on the relative subregional contributions to the seasonal cycle of China's GPP.

## 4. Discussion

### 4.1 Understanding the contribution of LULCC

The TBMs used in this study relied on LULCC data by combining a static satellite-based land cover product (Jung *et al*., 2006) with time-varying land use harmonization version 1 (LUH1) data (Hurtt *et al*., 2011). Based on this dataset, time series of different vegetation cover types over China and the nine sub-regions were developed and are presented in Fig. 7 (solid lines). Crop areas showed

a persistent increase during the past three decades (from 13% to 18%), whereas forest areas were shrinking (from 25% to 20%). Grassland areas showed a slight increase in the 1990s and then changed little during the past two decades. These changes induced a decrease in the mean values of China's





GPP (Table 1a). LULCC in China showed significant spatial variations. For example, changes in grassland occurred mostly in Inner Mongolia (R2, Fig. 7c). Cropland expansion was widely distributed across China, but at different rates in each sub-region. As for forest land, the largest loss occurred over northern China (Fig. 7e) and parts of southern China (Figs. 7f, 7h, 7i, and 7j).

LULCC in China from the LUH1 product used in this study showed some differences from previous studies. For example, Liu and Tian (2010) reconstructed an LULCC dataset for China using high resolution satellite and historical survey data and found that LULCC in China during 1980–2005 was characterized by shrinking cropland and expanding urban and forest areas. Chen (2007) also reported a similar trend of shrinking cropland in China during 1977–2003 and attributed it to

urbanization. Several studies have reported an increase in forest area after 1980 (Fang *et al*., 2001; Houghton and Hackler, 2003; Song and Deng, 2017), which was mainly due to new plantings to protect the environment (Wang *et al*., 2004). To assess the reliability of LULCC data used in this study, we compared them with the China Land Use/Cover Dataset (CLUD) (Liu, et al., 2003, 2005, 2010, 2014; Kuang et al., 2016), which was generated using two satellite datasets: the LandsatTM/ETM+ and HJ-

1A/1B images from the China Centre for Resources Satellite Data and Application (http://www.cresda.com/). The CLUD is a national high-resolution database (1 km) and contains the longest time-series dataset available for LULCC in China (Kuang et al., 2016). Its classification system includes six classes (woodland, cultivated land, grassland, water bodies, built-up land and unused land) and 25 subclasses (Liu et al., 2005; Zhang et al., 2014). The accuracy assessments for the CLUD have

been addressed in previous studies (Liu et al., 2003, 2005, 2010, 2014; Kuang et al., 2013, 2016). Based on the CLUD, the maps of main vegetation types in 1990, 1995, 2000, and 2010 were used here and their temporal changes in China and nine sub-regions are shown in Fig. 7 (dashed lines with dots). It is noted that CLUD is not available before 1990. In general, the LULCC data used in the MsTMIP agree well with the CLUD between 1990 and 2005, except some discrepancies in 2010. Compared to

that in 2000, the CLUD showed a slight increase in forest (from 20% to 22%) and shrinking cropland (from 31% to 21%) and grassland (from 20% to 14%) in 2010 for the whole China. The decrease in cropland was mainly from R1 (Fig. 7b), R4 (Fig. 7e), R5 (Fig. 7f) and R7 (Fig. 7h), while the changes in grassland occurred mostly in R2 (Fig. 7c), R3 (Fig. 7d), and R6 (Fig. 7g).

The uncertainties in the LULCC dataset could influence its contribution to terrestrial carbon

fluxes. In upcoming revisions to LUH1, the new LUH2 product (http://luh.umd.edu/data.shtml) includes updated inputs, higher spatial resolution, more detailed land use transitions, and the addition of important agricultural management layers. Moreover, forest cover gross transitions are now constrained by remote-sensing information and have generally been re-estimated. Therefore, future studies are expected to compare the potential effect on GPP with the new product.



**4.2 Uncertainties in simulating GPP in China**

Despite growing efforts to quantify GPP, current TBM simulations still have large uncertainties. Each TBM has different parameterizations, which led to its own bias, and the ensemble mean of multi-model simulations may reduce the bias in GPP (Ito *et al.*, 2016; Chen *et al.*, 2017). Therefore, this study did not focus on comparisons among the 12 model simulations.

The multi-model mean of the twelve MsTMIP models (SG3) for total China GPP was 7.4 Pg C yr$^{-1}$, which was slightly higher than the MTE estimate (7.0 Pg C yr$^{-1}$). The results in this study also showed some differences with previous studies. For example, China GPP estimates based on the Eddy-Covariance Light Use Efficiency model were 5.38 (Yuan *et al.*, 2010), 5.55 (Cai *et al.*, 2010), and 6.04 Pg C yr$^{-1}$ (Li *et al.*, 2010) respectively, which were more than 20% lower than in this study. Yao *et al.* (2018) developed a new GPP product for China with higher spatial resolution (0.1°) based on a machine-learning algorithm using more eddy flux observations than the MTE. They found that the annual GPP of China was 6.62 ± 0.23 Pg C yr$^{-1}$ during 1982–2015. In contrast, the ensemble mean of nine TBMs produced a higher estimate of 7.85 Pg C yr$^{-1}$ (Yao *et al.*, 2018). In addition, two newly published studies also generated high estimates of total annual GPP: 7.85 Pg C yr$^{-1}$ for 2001–2010 by multiple regression (Zhu *et al.*, 2014) and 7.81 Pg C yr$^{-1}$ for 2000–2015 using support vector regression (Ichii *et al.*, 2017). Unlike the discrepancies in the magnitude of annual mean China GPP, the trend in this study is very similar to that of Yao *et al.* (2018), with a positive value of 0.02 Pg C yr$^{-2}$ ($p < 0.05$).

In this study, MsTMIP and MTE were found to show some discrepancies in the IAV and trends of GPP. For example, the trend of MsTMIP is about twice of that derived from the MTE data (Fig. 3). The reason for the differences can be explained through the following two aspects. First of all, uncertainties in meteorological forcing dataset, model structure and parameterization can lead to large biases in simulating the spatial-temporal patterns of GPP although this could be reduced by ensemble simulations from MsTMIP. Secondly, although data-oriented GPP product (e.g. the MTE) has been used as the reference data to evaluate the TBM simulations (Piao et al., 2012, 2013; Jia et al., 2018; Yao et al., 2018), previous studies found that MTE data may underestimate the IAV and trends (Jung et al., 2011; Piao et al., 2013). It may be due to the potential biases caused by "spatial gradients extrapolation to temporal interannual gradients" (Reichstein et al., 2007; Jung et al., 2009; Piao et al., 2013; Yao et al., 2018), and leaving out some cumulative effects like soil moisture (Jung et al., 2007). In addition, most of the stations used by the MTE data only had short measurements period (Yao et al., 2018), which may affect the estimations of long-term temporal variations in GPP (e.g., IAV, trend).



It is noted that the latest version of the MTE data agreed with TBM simulations well (Jung et al., 2017), which will be compared with the GPP estimations over China from MsTMIP in our future work.

## 5. Conclusions

In this study, a multi-model analysis using twelve MsTMIP-based models was used to investigate the relative contributions of climate change and anthropogenic activities to interannual and seasonal variations in China's GPP. In addition, this study examined subregional contributions to temporal changes in China's total GPP. Ensemble simulations from MsTMIP were compared with an independent upscaling GPP product (Jung *et al*., 2011) and with flux tower-based GPP observations in China.

The simulated GPP for China from the 12 MsTMIP models, driven by common climate forcing, LULCC, and $CO_2$ data, was $7.4\pm1.8$ Pg C $yr^{-1}$, which agreed well with independent MTE data set (7.1 Pg C $yr^{-1}$). In general, climate was the dominant control factor for the trends, interannual variation, and seasonality of China's GPP. The overall rise in $CO_2$ enhanced plant photosynthesis and thus increased total China GPP, with increasing annual mean and interannual variability, especially in northeastern and southern China where vegetation is dense.

Note that existing model estimates of GPP from state-of-the-art TBMs vary widely and still have large uncertainties driven by biases in environmental driver data and unrealistic assumptions in model parameterizations and parameters (Friedlingstein *et al*., 2006; Huntzinger *et al*., 2012). The multi-model ensemble strategy is a means to address model structural uncertainty by synthesizing outcomes from multiple models representing different parameterizations of underlying biogeophysical and biogeochemical processes, and has been demonstrated to offer better predictability (Hagedorn *et al*., 2005). However, there are some missing factors that are not considered in this study. One is that the interaction between LULCC and elevated $CO_2$ was not completely separated in this study. For example, deforestation under the background of raising $CO_2$ induces higher emissions because $CO_2$ fertilization leads to an increase in terrestrial carbon storage, but higher $CO_2$ concentrations also cause a stronger regrowth (Houghton *et al*., 2012). Moreover, the uncertainty in LULCC data sets remains a serious challenge today. More satellite data with higher spatial resolution are expected to reduce this uncertainty.





**Acknowledgments**

This research was supported by the National Key R & D Program of China (2016YFA0600203), the Key Research Program of Frontier Sciences, CAS (QYZDY-SSW-DQC012), the National Natural Science Foundation of China (41575096, 41830967), and the Key Lab of Guangdong for Utilization 5 of Remote Sensing and Geographical Information System, Guangzhou Institute of Geography (2017B030314138). Finalized MsTMIP data products are archived at the ORNL DAAC (http://daac.ornl.gov). The MTE data were downloaded freely from the Max Planck Institute for Biogeochemistry (https://www.bgc-jena.mpg.de/). We acknowledge the MsTMIP modelers, including Dr. Maoyi Huang from Pacific Northwest National Laboratory, Dr. Shushi Peng from Peking 10 University, Dr. Joshus B Fisher from California Institute of Technology, and Dr. Daniel M Ricciuto from Oak Ridge National Laboratory, for contributing model output used in this work. We also thank Dr. Wenhui Kuang for providing us the China Land Use/Cover Dataset (CLUD).




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



**Tables**

**Table 1.** China and regional GPP from the MTE and the ensemble mean of the twelve MsTMIP models for three configurations (SG1, SG2, and SG3): (a) mean values (MEAN), (b) interannual variability (IAV), (c) seasonal-cycle amplitude (SCA).

(a) MEAN (units Pg C yr$^{-1}$)

|  | SG1 | SG2 | SG3 | MTE |
|---|---|---|---|---|
| China | 6.9 | 6.7 | 7.4 | 7.0 |
| R1 | 0.8 | 0.8 | 0.9 | 0.8 |
| R2 | 0.4 | 0.4 | 0.4 | 0.4 |
| R3 | 0.3 | 0.3 | 0.3 | 0.3 |
| R4 | 0.7 | 0.7 | 0.8 | 0.7 |
| R5 | 0.8 | 0.7 | 0.8 | 0.7 |
| R6 | 0.5 | 0.4 | 0.5 | 0.5 |
| R7 | 1.2 | 1.2 | 1.3 | 1.2 |
| R8 | 1.0 | 0.9 | 1.0 | 1.0 |
| R9 | 1.5 | 1.4 | 1.5 | 1.4 |

(b) IAV (unit Pg C yr$^{-1}$)

|  | SG1 | SG2 | SG3 | MTE |
|---|---|---|---|---|
| China | 0.099 | 0.092 | 0.105 | 0.157 |
| R1 | 0.030 | 0.033 | 0.030 | 0.029 |
| R2 | 0.024 | 0.021 | 0.023 | 0.025 |
| R3 | 0.010 | 0.012 | 0.010 | 0.015 |
| R4 | 0.030 | 0.029 | 0.033 | 0.048 |
| R5 | 0.025 | 0.022 | 0.024 | 0.020 |
| R6 | 0.018 | 0.016 | 0.018 | 0.014 |
| R7 | 0.034 | 0.032 | 0.033 | 0.025 |
| R8 | 0.030 | 0.031 | 0.031 | 0.019 |
| R9 | 0.031 | 0.029 | 0.032 | 0.029 |

(c) SCA (unit Pg C yr$^{-1}$)

|  | SG1 | SG2 | SG3 | MTE |
|---|---|---|---|---|
| China | 11.1 | 10.8 | 12.3 | 13.6 |





| | | | |
|----|-----|-----|-----|
| R1 | 2.3 | 2.2 | 2.6 | 2.8 |
| R2 | 1.1 | 1.0 | 1.2 | 1.3 |
| R3 | 0.7 | 0.7 | 0.8 | 1.1 |
| R4 | 1.4 | 1.4 | 1.6 | 2.2 |
| R5 | 0.9 | 0.9 | 1.1 | 1.2 |
| R6 | 1.2 | 1.0 | 1.2 | 1.1 |
| R7 | 1.3 | 1.3 | 1.5 | 1.5 |
| R8 | 0.8 | 0.8 | 0.9 | 1.0 |
| R9 | 1.8 | 1.7 | 1.9 | 2.2 |

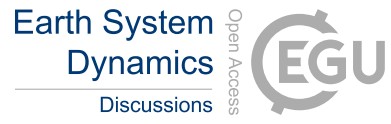

**Figures**

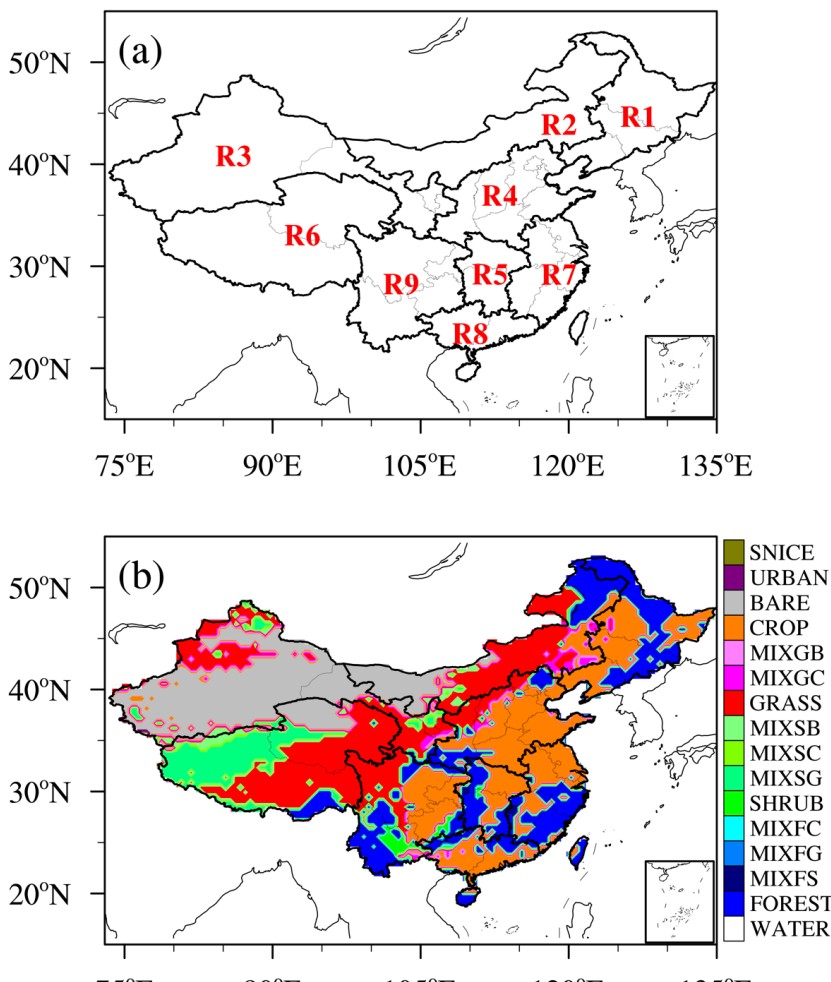

**Figure 1.** Spatial distributions of (a) nine sub-regions (R1−R9) in China; and (b) present plant functional types (PFT) used in SG1 simulations. R1: northeastern China (Heilongjiang, Jilin, Liaoning); R2: Inner Mongolia; R3: northwestern China (Gansu, Ningxia, Xinjiang); R4: northern China (Beijing, Hebei, Henan, Shandong, Shanxi, Shaanxi, Tianjin); R5: central China (Hubei, Hunan); R6: Tibetan Plateau (Qinghai, Tibet); R7: southeastern China (Anhui, Fujian, Jiangsu, Jiangxi, Shanghai, Taiwan, Zhejiang); R8: southern China (Guangdong, Guangxi, Hainan, Hong Kong, Macao); and R9: southwestern China (Guizhou, Sichuan, Yunnan, Chongqing). SNICE is snow and ice, BARE is bare soil, MIXFS is mixed forest and shrubs, MIXFG is mixed forest and grass, MIXFC is mixed forest and crops, MIXSG is mixed shrubs and grass, MIXSC is mixed shrubs and crops, MIXSG is mixed



shrubs and grass, MIXSB is mixed shrubs and bare soil, MIXGC is mixed grass and crops, and MIXGB is mixed grass and bare soil.





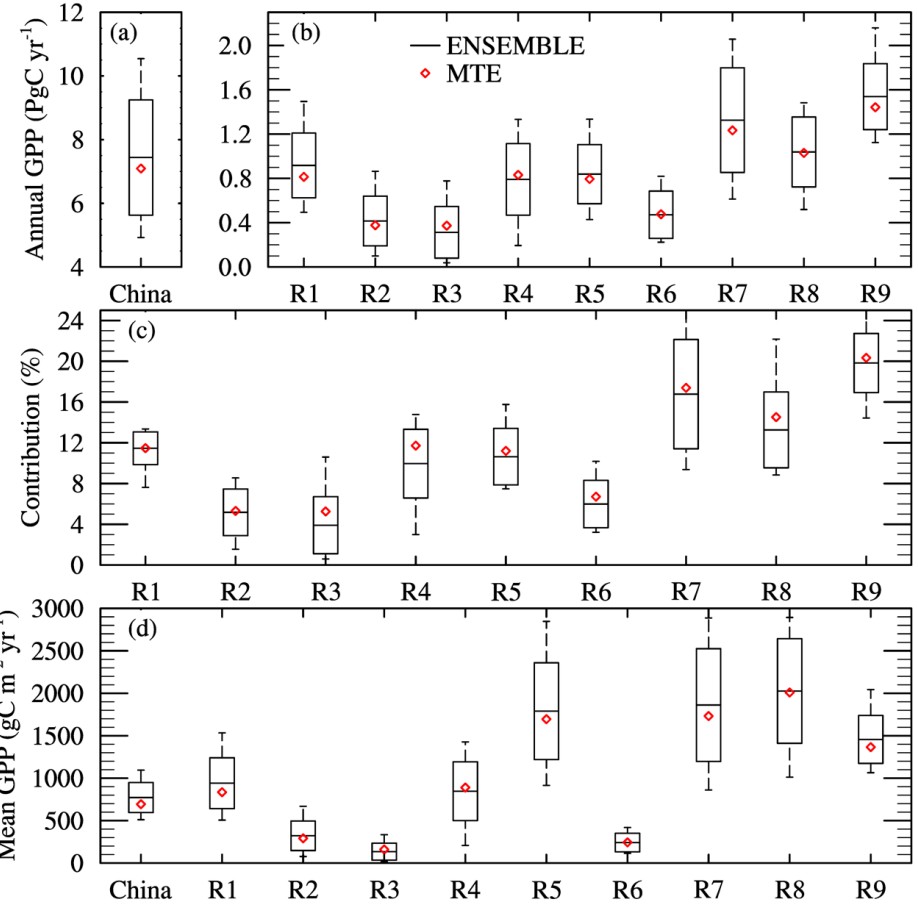

**Figure 2.** Annual mean GPP from (a) China and (b) each sub-region; (c) regional contributions to China GPP; (d) annual mean GPP per unit square meters. Horizontal lines at top, middle, bottom in the boxplots represent the maximum, ensemble mean, and minimum of multi-model simulations respectively, whereas the box indicates one standard deviation. All the results in this figure are from 1981–2010 for the MsTMIP SG3 simulation and 1982–2010 for the MTE. Regional abbreviations used on the x-axes are defined in Fig. 1a.





**Figure 3.** Interannual variations in GPP of China and each sub-region from the MTE (black) and the ensemble mean of the 12 MsTMIP models: SG1 (blue), SG2 (red), SG3 (purple). The anomalies of GPP were calculated as the difference between annual GPP and the long-term mean between 1981 and 2010 (MTE is 1982-2010). The numbers located at the top of each figure indicate the linear trends of

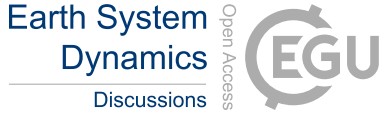

SG1 (blue), SG2 (red), SG3 (purple), and MTE (black) with units of Tg C yr$^{-2}$ (1 Tg C=0.001 Pg C).

$^{*}$ and $^{**}$ indicate the trend is significant with $p < 0.1$ and $p < 0.05$ respectively.



Earth System
Dynamics
Discussions
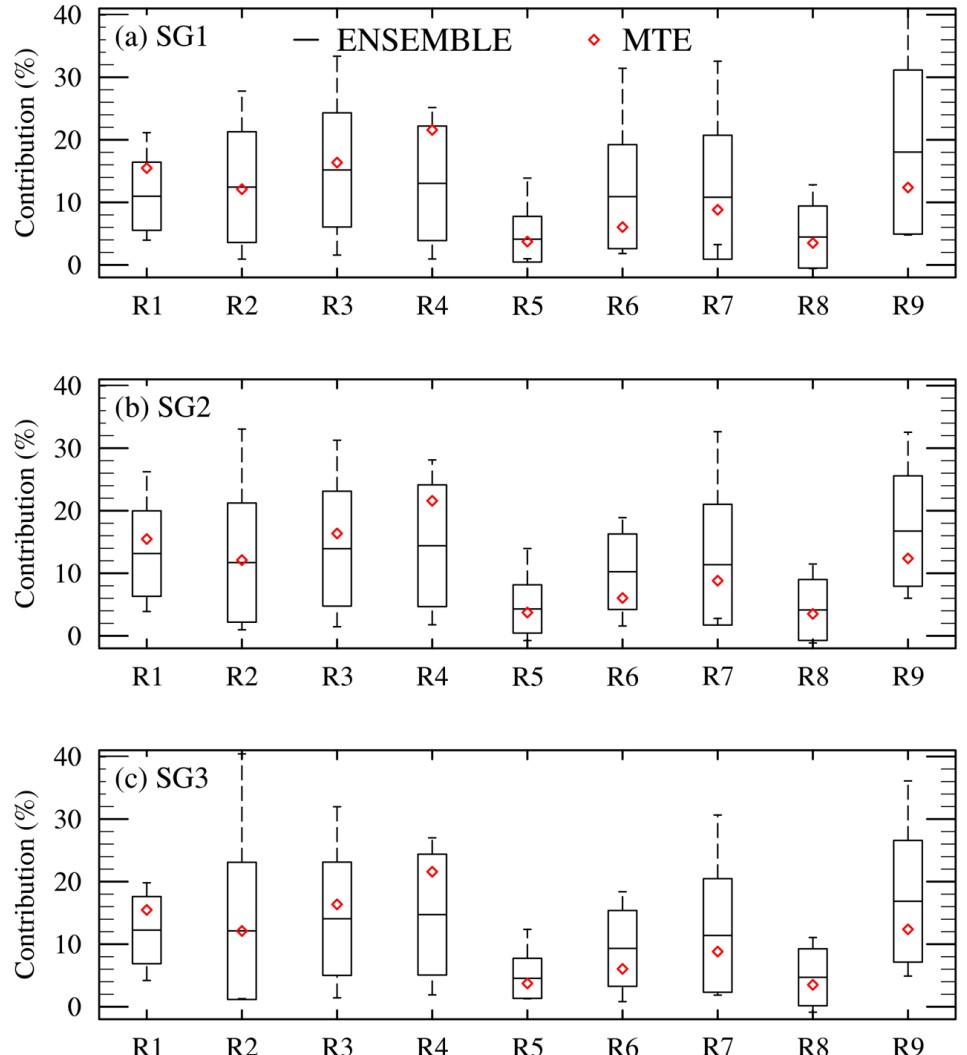

**Figure 4.** The relative contributions of each sub-region to the interannual variability (IAV) of China's
GPP. Boxplots indicate the distributions of the 12 MsTMIP models. Horizontal lines at top, middle,
and bottom in the boxplots represent the maximum, ensemble mean, and minimum of multi-model
5   simulations respectively, whereas the box indicates one standard deviation. All the results in this figure
are from 1981–2010 for the MsTMIP models and 1982–2010 for the MTE.





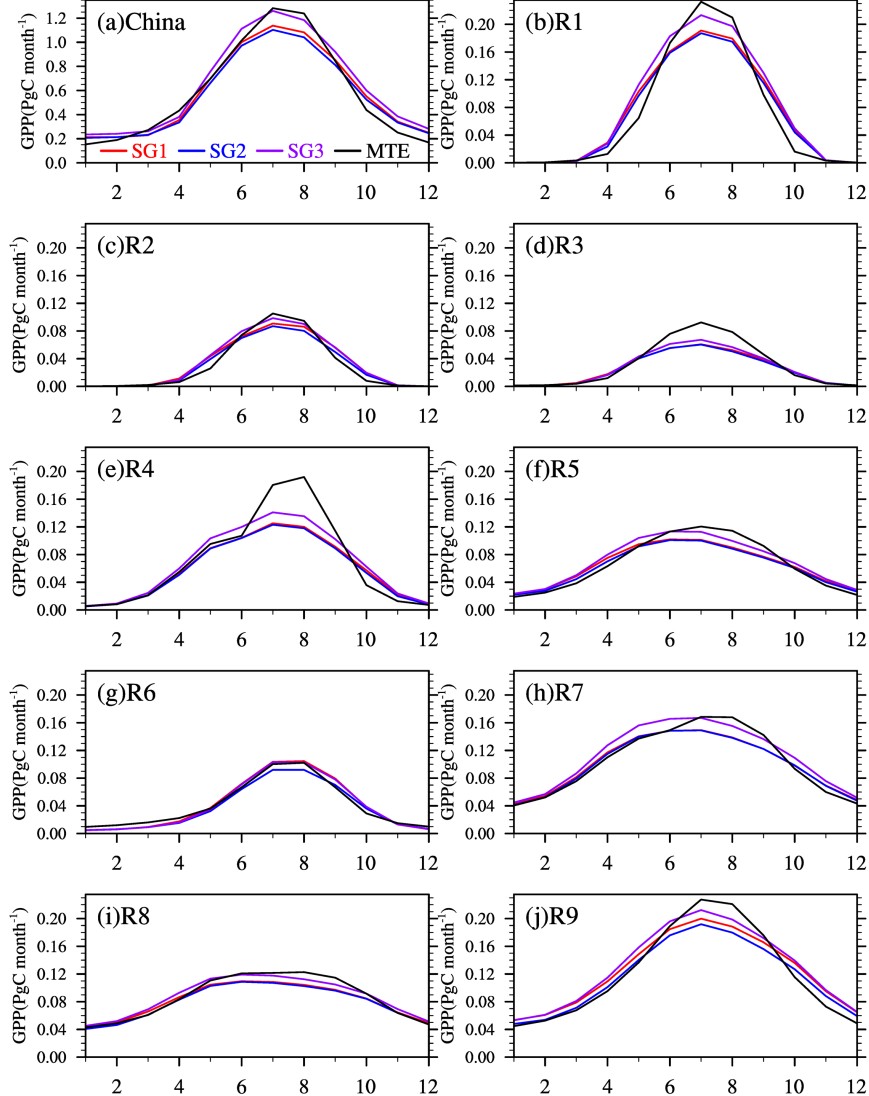

**Figure 5.** Mean seasonal variations in GPP of China and each sub-region from the ensemble mean of the twelve MsTMIP models for the three simulations (SG1, SG2, and SG3) between 1981 and 2010. Note that the MTE is between 1982 and 2010.



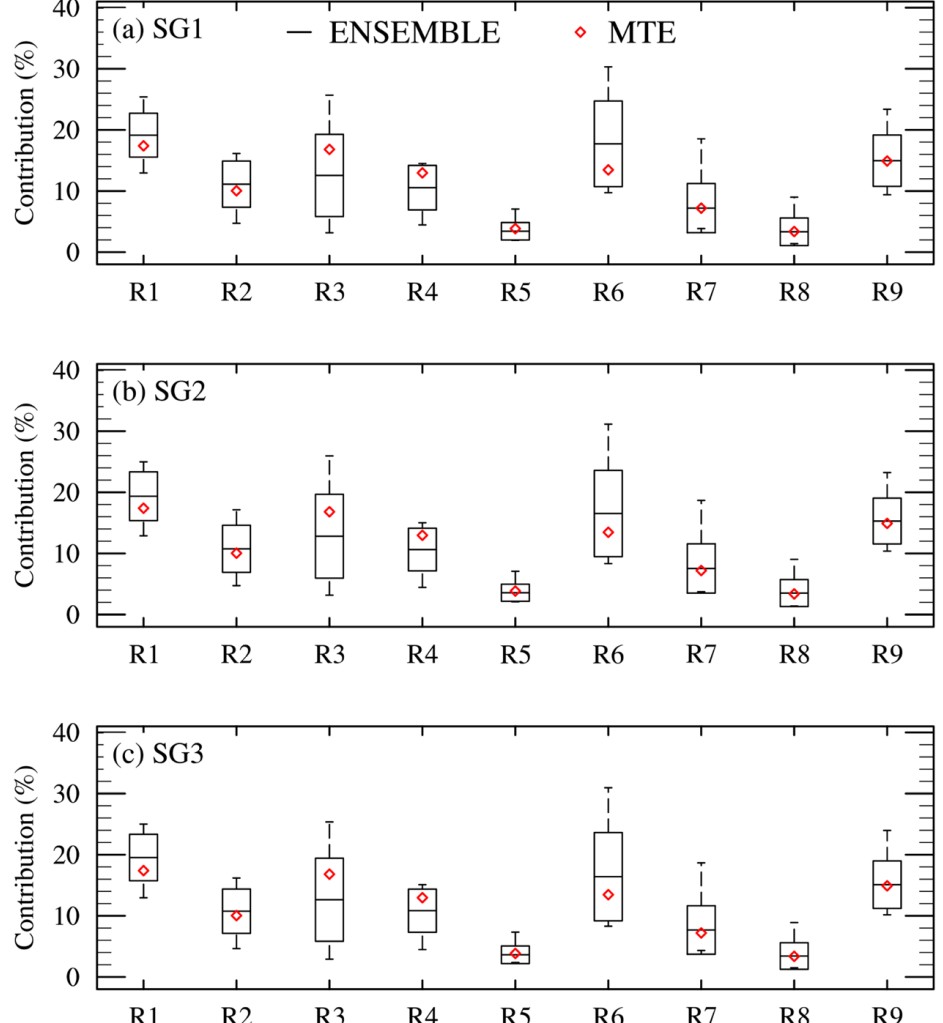

**Figure 6.** The same as Fig. 3, but for the regional contributions to the seasonality of China's GPP.





**Figure 7.** Interannual changes of vegetation types over China and nine sub-regions between 1981 and 2010 from the MsTMIP (solid lines) and the China Land Use/Cover Dataset (dashed lines with dots).