# Peer review of "Impacts of land-use change and elevated CO2 on the interannual variations and seasonal cycles of gross primary productivity in China"

_Earth System Dynamics, 2019_

## Referee Comment (RC1) · Anonymous Referee #1 · 16 Jun 2019

Binghao Jia et al. investigates the effect of CO2, climate and land use change on the inter-annual variation and seasonal cycle of gross primary production (GPP) in China using 12 terrestrial biosphere models and observation driven data. Their main finding, in general, is that climate was the dominant control factor during 1981-2010 for the trends, inter-annual variations, and seasonality of China's GPP. A rise in CO2 increased GPP in China with increased inter-annual variability especially in the places where vegetation is dense.

→ I like the way authors choose to analyze the GPP data and perform the statistical tests from 12 models along with an observation-based estimate.

[Figure]

→ The nonparametric method is used to test statistical significance.

→ Figures are carefully chosen to communicate the essential results.

→ References are appropriately cited

I find the paper is well written and the presentation is excellent. I thus recommend this paper for publication once the following minor issues are addressed.

1) I am a bit surprised that China Land Use/Cover Dataset in Fig.7 shows a decrease in cropland areas at end of the period (1982-2010) and also many satellite-based studies (you have also listed many peer-reviewed) over China shows large afforestation but Terrestrial biosphere models show an increase in cropland areas?? Then how come observation-based estimate of GPP you have shown agrees very well with the model simulated GPP? Could you please clarify this more clearly in the text?

2) I suggest moving supplementary Figure S1 to main Figure 2. This is an important figure and also you are discussing this right at the start of the results section and I think this should be moved. There is an inset figure in each panel of this figure? What is it? You don't discuss. Explain what is it otherwise remove! Also, in the caption please be clear that the results shown in Fig.S1 (also many other figure captions) are the average of 28 years or what?

3) Page 5, two lines above the line 25: I suspect that Fig.2a instead of Fig.1a. Also, the GPP range you mentioned 4.9 to 9.2 PgC/yr, but I see Fig.2a maximum value in the boxplot is more than 10PgC/yr.

4) In general, throughout the text, it would be convenient for the readers if you mention also the abbreviation for the regions (R1, R2, etc..). For eg. at Page 5, two lines below line 25: southeastern China (1.3 PgC/yr, R7) and (1.5 PgC/yr, R9)? I suppose 1.5 PgC/yr corresponds to R9.

5) Why MTE abbreviation for the machine learning algorithm?? not sure how you have chosen MTE?

[Figure]

6) In Figures 3 & 7 legends should be at the top left/right panel (eg. Fig.3a or 3b), to avoid wondering which color is what for a while. Readers usually start looking at the first panel of the figure before going to the bottom panels.

7) At page 7, near line 10: why some discrepancies between SG3 and MTE over northeast China, southeastern China and east parts of southwestern China? Worth explaining there!

---

## Referee Comment (RC2) · Anonymous Referee #2 · 1 Sep 2019

The paper has discussed total GPP and its regional distribution in China from 1981 to 2010 using results from 12 terrestrial biosphere models. Effect of LULCC and atmospheric CO2 levels on GPP in China has also been studied by analysing results from different experiments that were well-described in the text. Overall, the paper is comprehensive in terms of understanding the effect of LULCC and CO2 on GPP for China for recent years. Validation of the results, the use of ensemble mean for the purpose of this study and representation of the figures is appropriate. Congratulations to the authors for coming up with a detailed study. The manuscript is well-written overall. However, I have some issues as described in detail below:

[Figure]

Major comments:

1. Identification of gaps in literature has not been done adequately in the Introduction section. Page 3, line 16 "However, few studies have adequately explored the impacts of climate change, atmospheric $CO_2$ concentration, and LULCC to interannual and seasonal variations of GPP in China". If there are already studies that have studied these impacts, they should be cited here and effects of LULCC and $CO_2$ on GPP as estimated in this study should be compared with these studies in the later sections.

2. Page 4, line 19, explanation of the term MTE is not very clear. This should be made clear before MTE is used to represent the dataset in the rest of the paper from this point on.

3. Figure 1 (on page 23) shows 16 different kinds of vegetation types in the legend but only the major ones are visible in the plot. To make the plot readable, similar vegetation types like MIXSB, MIXSC, MIXSG, SHRUB should be merged since they are anyway not much distinguishable in the plot.

4. Page 5, Section: 3.1. Since this section starts with the discussion of results presented in Fig. S1 and has an entire paragraph on this figure, the figure should be moved to the main text.

5. There is a lot of mismatch between the region references in terms of region names and regions numbers in the Results section. For instance: a. Page 7, line 13, "central China and northern China" should rather be "northern China (R4) and northwestern China (R5)", as per the numbers represented in figure 4. b. Page 7, line 32, "in summer over southeastern China (Fig. 5j)". 5j corresponds to R9 and as per fig. 1, R9 is southwestern China, not southeastern China. To avoid this confusion in region names and region numbers, I would strongly recommend the authors to double check the text in the sections of Results and Discussions, and to use region numbers along with region names in these sections to that the text explanation can be verified easily with the figures.

6. Page 11, line 12 and Page 1, line 34: A strong concluding statement has been made about how climate is the dominant control factor of annual trends, IAV and seasonality of China's GPP, without much analysis of results in this context in the Results section. Some analysis of trends coming from SG1 case should be included in the results section before making this statement, specifically since the paper has focussed mostly on LULCC and CO2 effects, and there are not many remarks on impact of climate in the paper.

7. The implications of this study and application of the results are not adequately emphasised. The authors are suggested to add some information on how this work his valuable, specifically considering how understanding of the effects of LULCC and CO2 on GPP can help in comprehensive scenario of things and decision making.

Other issues to be considered:

1. There is no mention of the study period in the abstract so it is not clear for which years are the results mentioned in this section applicable for.

2. The phrase "independent upscaling GPP estimate" in the abstract does not give any idea of the dataset being talked about and hence should be either modified or eliminated from this section.

3. The usage of a few words and sentence formation in the text is questionable in some places, for instance: a. Page 2, lines 25 and 27: "60% of the uptake by terrestrial ecosystem was due to raising(?) atmospheric CO2" and "It suggests that the impact of raising(?) CO2 on land carbon sink may be a negative feedback to future climate". b. Page 4, line 14: "The simulated monthly GPP from these 12 models was conducted(?) for the period of 1981–2010." The authors are suggested to re-check these typos and small errors.

4. Table S1 (mentioned on Page 4, line 10) only has all "O" under columns SG1, SG2 and SG3 for all models, check attached file. I am not sure what purpose the table is

serving apart from citing references for each model description. This table can either be improved or deleted.

5. Page 5, line 23, Fig. 1a.(?). This seems to be a typo and Fig. 2a. should be mentioned here.

6. Figure 7 has comparison of LUH1 data with CLUD for major vegetation types. Clearly, there is a mismatch in the recent trends of both datasets, more specifically from year 2000 to 2010. This difference is intriguing but since the figure does not represent 100% land cover of China, there is missing information here. For instance, the sum of major vegetation types shown in fig. 7a represents $\sim$ 60% of area for CLUD for 2010 and $\sim$ 80% of land cover for LUH1 for 2010. I would suggest this figure to be modified to account for 100% area of China so that the entire land cover distribution and the transitions can be accounted for.

---

## Author Comment (AC1) · 16 Sep 2019

We thank the reviewer (Referee) for the constructive comments and suggestions, which are in the text below. Our itemized response is followed.

Background

Binghao Jia et al. investigates the effect of CO2, climate and land use change on the inter-annual variation and seasonal cycle of gross primary production (GPP) in China using 12 terrestrial biosphere models and observation driven data. Their main finding, in general, is that climate was the dominant control factor during 1981-2010

for the trends, inter-annual variations, and seasonality of China's GPP. A rise in CO2 increased GPP in China with increased inter-annual variability especially in the places where vegetation is dense.

- I like the way authors choose to analyze the GPP data and perform the statistical tests from 12 models along with an observation-based estimate.

- The nonparametric method is used to test statistical significance.

- Figures are carefully chosen to communicate the essential results.

- References are appropriately cited.

I find the paper is well written and the presentation is excellent. I thus recommend this paper for publication once the following minor issues are addressed.

Comments:

1. I am a bit surprised that China Land Use/Cover Dataset in Fig. 7 shows a decrease in cropland areas at end of the period (1982-2010) and also many satellite-based studies (you have also listed many peer-reviewed) over China shows large afforestation but Terrestrial biosphere models show an increase in cropland areas?? Then how come observation-based estimate of GPP you have shown agrees very well with the model simulated GPP? Could you please clarify this more clearly in the text?

Response: Based on the comments, we added some discussions about the effect of LULCC on the comparison between model simulated GPP and observation-based estimates.

(1) The LULCC data used in the TBMs from MsTMIP were generated by combining a static satellite-based land cover product (Jung et al., 2006) with time-varying land use harmonization version 1 (LUH1) data (Hurtt et al., 2011). The satellite-based LULCC data set, named the China Land Use/Cover Dataset (CLUD) (Liu, et al., 2003, 2005, 2010, 2014; Kuang et al., 2016), was generated using two satellite datasets: the LandsatTM/ETM+ and HJ-1A/1B images from the China Centre for Resources Satellite Data and Application (http://www.cresda.com/). In general, the LULCC data used in the MsT-MIP agree well with the CLUD between 1990 and 2005, except some discrepancies in 2010. Compared to that in 2000, the CLUD showed a slight increase in forest (from 20% to 22%) and shrinking cropland (from 31% to 21%) and grassland (from 20% to 14%) in 2010 for the whole China. The decrease in cropland is mainly from R1 (Fig. 8b), R4 (Fig. 8e), R5 (Fig. 8f) and R7 (Fig. 8h). The main cause may be related with the expanding urban and forest areas. For example, rapid urbanization over eastern China induce the decrease in the cropland. In addition, due to the government-issued policies for protecting the environment, many national afforestation and reforestation projects have been implemented in China, which lead to the conversion of cropland to forest. In contrast, LULCC used in the TBMs show an increase in the cropland.

(2) The differences in the LULCC data indeed affect the model simulated GPP. For example, the MTE GPP products show a significantly increasing trend after 2005 over R4 (Fig. 4e), R5 (Fig. 4f) and R7 (Fig. 4h) while some underestimations can be found for model simulated GPP. This may be related with the discrepancies in the LULCC data sets over these areas. Please see Page 10 (Lines 2−4).

2. I suggest moving supplementary Figure S1 to main Figure 2. This is an important figure and also you are discussing this right at the start of the results section and I think this should be moved. There is an inset figure in each panel of this figure? What is it? You don't discuss. Explain what is it otherwise remove! Also, in the caption please be clear that the results shown in Fig.S1 (also many other figure captions) are the average of 28 years or what?

Response: (1) Based on the suggestions, old Figure S1 has been moved to the revised manuscript to be the new Fig. 2. Please see Page 25. (2) The inset figure in the bottom-right corner of Fig. 1, new Fig. 2, new Fig S2 represents the boundary of China. We have removed these inset figures based on your suggestions. (3) The results shown in the new Fig. 2 (old Fig. S1) are the averages of 30 years for MsTMIP models
(1981−2010) and 29 years for the MTE (1982−2010), respectively. We have revised the captions of all the relative figures (Figs. 2, 3, 5, 6) according to the comments. Please see Pages 24−25, 27−28.

3. Page 5, two lines above the line 25: I suspect that Fig.2a instead of Fig.1a. Also, the GPP range you mentioned 4.9 to 9.2 PgC/yr, but I see Fig.2a maximum value in the boxplot is more than 10PgC/yr.

Response: These sentences have been revised based on the suggestions. It should be Fig. 3a (old Fig. 2a). The total China GPP ranges from 4.9 (DLEM) to 10.5 (GTEC) Pg C yr−1. Please see Page 5 (Lines 25−26).

4. In general, throughout the text, it would be convenient for the readers if you mention also the abbreviation for the regions (R1, R2, etc..). For eg. at Page 5, two lines below line 25: southeastern China (1.3 PgC/yr, R7) and (1.5 PgC/yr, R9)? I suppose 1.5 PgC/yr corresponds to R9.

Response: Based on the suggestion, all the abbreviations have been added into the revised manuscript. For this example, it has been revised to be: "The regional sum of GPP in southwestern China from the ENSEMBLE (Fig. 3b) was the highest among all nine regions (1.5 Pg C yr−1, R9), followed by southeastern China (1.3 Pg C yr−1, R7) and southern China (1.0 Pg C yr−1, R8)". Please see Page 5 (Lines 29−31). The other revisions can be found in the revised manuscript.

5. Why MTE abbreviation for the machine learning algorithm?? not sure how you have chosen MTE?

Response: Based on the comments, we added more descriptions about the MTE. The observation-based GPP product were generated using the machine-learning algorithm, model tree ensembles (MTE). Therefore, we used the "MTE" to represent this product. Please see Page 4 (Lines 18−28): "This study used an observation-driven global monthly gridded GPP product derived from FLUXNET measurements by sta-
tistical upscaling with the machine-learning algorithm, model tree ensembles (Jung et al., 2009, 2011) (hereafter referred to as MTE). The MTE statistical model consisting of a set of regression trees was firstly trained using site‐level explanatory variables and GPP estimations from eddy flux tower measurements. These explanatory variables covered climate and biophysical variables such as vegetation types, temperature, precipitation, radiation, and satellite-derived fraction of absorbed photosynthetic active radiation. Then the MTE GPP product was generated through applying the trained regression trees for global upscaling using gridded data sets of the same explanatory variables. It has a spatial resolution of $0.5° \times 0.5°$ and is available between 1982 and 2011. The uncertainty of the MTE data is $\sim$46 g C m$-$2 yr$-$1 (5%), which was calculated using the standard deviation of the 25 model tree ensembles (Jung et al., 2011)".

6. In Figures 3 & 7 legends should be at the top left/right panel (eg. Fig.3a or 3b), to avoid wondering which color is what for a while. Readers usually start looking at the first panel of the figure before going to the bottom panels.

Response: Based on the suggestions, we have moved the legends to the top-left panel of the two figures. Please see the new Fig. 4 on Page 26 and new Fig. 8 on Page 30.

7. At page 7, near line 10: why some discrepancies between SG3 and MTE over northeast China, southeastern China and east parts of southwestern China? Worth explaining there!

Response: Based on this comment, we added some explanations about the differences of GPP trend between SG3 and MTE over these regions. Since MsTMIP SG3 simulations and MTE product were generated using different methods, we then compared them with another GPP data set from Yao et al. (2018) (hereafter YAO, Fig. 4a in that paper). YAO is a new GPP product for China with higher spatial resolution ($0.1°$) based on the same machine-learning algorithm with the MTE product, but it uses more eddy flux observations (40 flux sites). It is found that SG3 from MsTMIP shows similar trends with YAO over R1 and east parts of R7. In contrast, MTE shows the same increasing trends with YAO over east parts of R9. It suggests that both model simulations from MsTMIP and MTE GPP product shows certain uncertainties in the GPP trend over some areas of China, which needs more observations to evaluate the GPP trend in future work. Please see Page 7 (Lines 12−18).

References

[1] Hurtt, G. C., Chini, L., Frolking, S., Betts, R., Edmonds, J., Feddema, J., Fisher, G., Goldewijk, K. K., Hibbard, K., Houghton, R., Janetos, A., Jones, C., Kinderman, G., Konoshita, T., Riahi, K., Shevliakova, E., Smith, S. J., Stefest, E., Thomson, A. M., Thornton, P., van Vuuren, D., and Wang, Y.: Harmonization of land-use scenarios for the period 1500–2100: 600 years of global gridded annual land-use transitions, wood harvest, and resulting secondary lands, Clim. Change, 109, 117–161, doi:10.1007/s10584-011-0153-2, 2011.

[2] Jung, M., Henkel, K., Herold, M., and Churkina, G.: Exploiting synergies of global land cover products for carbon cycle modeling, Remote Sens. Environ., 101, 534–553, doi:10.1016/j.rse.2006.01.020, 2006.

[3] Liu, J., Kuang, W., Zhang, Z., Xu, X., Qin, Y., Ning, J., Zhou, W., Zhang, S., Li, R., Yan, C., Wu, S., Shi, X., Jiang, N., Yu, D., Pan, X., and Chi, W.: Spatiotemporal characteristics, patterns, and causes of land-use changes in China since the late 1980s, J. Geogr. Sci., 24(2), 195–210, http://dx.doi.org/10.1007/s11442-014-1082-6, 2014.

[4] Liu, J., Liu, M., Tian, H., Zhuang, D., Zhang, Z., Zhang, W., Tang, X., and Deng, X.: Spatial and temporal patterns of China cropland during 1990–2000: An analysis based on Landsat TM data, Remote Sens. Environ., 98(4),442–456, 2005.

[5] Liu, J., Liu, M., Zhuang, D., Zhang, Z., and Deng, X.: Study on spatial pattern of land-use change in China during 1995–2000, Science in China Series D: Earth Sciences, 46(4), 373–384, 2003.

[Figure]

[6] Liu, J., Zhang, Z., Xu, X., Kuang, W., Zhou, W., Zhang, S., Li, R., Yan, C., Yu, D., Wu, S., and Jiang, N.: Spatial patterns and driving forces of land use change in China during the early 21st century, J. Geogr. Sci., 20(4), 483–494, http://dx.doi.org/10.1007/s11442-010-0483-4, 2010.

[7] Kuang, W., Liu, J., Dong, J., Chi, W., and Zhang, C.: The rapid and massive urban and industrial land expansions in China between 1990 and 2010: A CLUD-based analysis of their trajectories, patterns, and drivers, Landscape Urban Plan., 145, 21–33, 2016.

[8] Yao, Y., Wang, X., Li, Y., Wang, T., Shen, M., Du, M., He, H., Li, Y., Luo, W., Ma, M., Ma, Y., Tang, Y., Wang, H., Zhang, X., Zhang, Y., Zhao, L., Zhou, G., and Piao, S.: Spatiotemporal pattern of gross primary productivity and its covariation with climate in China over the last thirty years, Glob. Change Biol., 24, 184–196, 2018.

Please also note the supplement to this comment:
https://www.earth-syst-dynam-discuss.net/esd-2019-22/esd-2019-22-AC1-supplement.zip

---

## Author Comment (AC2) · 16 Sep 2019

We thank the reviewer (Referee #2) for the constructive comments and suggestions, which are in the text below. Our itemized response is followed.

Background

The paper has discussed total GPP and its regional distribution in China from 1981 to 2010 using results from 12 terrestrial biosphere models. Effect of LULCC and atmospheric CO2 levels on GPP in China has also been studied by analysing results from different experiments that were well-described in the text. Overall, the paper is

comprehensive in terms of understanding the effect of LULCC and CO2 on GPP for China for recent years. Validation of the results, the use of ensemble mean for the purpose of this study and representation of the figures is appropriate. Congratulations to the authors for coming up with a detailed study. The manuscript is well-written overall. However, I have some issues as described in detail below:

Major Comments:

1. Identification of gaps in literature has not been done adequately in the Introduction section. Page 3, line 16 "However, few studies have adequately explored the impacts of climate change, atmospheric CO2 concentration, and LULCC to interannual and seasonal variations of GPP in China". If there are already studies that have studied these impacts, they should be cited here and effects of LULCC and CO2 on GPP as estimated in this study should be compared with these studies in the later sections.

Response: Based on the comments, we revised this sentence by adding two relevant references. Moreover, another sentence was added to explain the differences between the two studies and the present work. Please see Page 3 (Lines 15−20): "However, few studies have adequately explored the impacts of climate change, atmospheric CO2 concentration, and LULCC to interannual and seasonal variations of GPP in China (Piao et al., 2013; Yao et al., 2018). These studies mainly focused on the climatic driver (temperature, precipitation, and solar radiation) of GPP interannual variations (Yao et al., 2018) and responses of GPP to climate variations and atmospheric CO2 concentration (Piao et al., 2013). But the quantitative contributions of these three factors on GPP in China are still unclear, which urgently needs to be addressed."

2. Page 4, line 19, explanation of the term MTE is not very clear. This should be made clear before MTE is used to represent the dataset in the rest of the paper from this point on?

Response: Based on this comment, we added a more detailed descriptions of MTE to this section. Please see Page 4 (Lines 18−26): "This study used an observation-

driven global monthly gridded GPP product derived from FLUXNET measurements by statistical upscaling with the machine-learning algorithm, model tree ensembles (Jung et al., 2009, 2011) (hereafter referred to as MTE). The MTE statistical model consisting of a set of regression trees was firstly trained using site-level explanatory variables and GPP estimations from eddy flux tower measurements. These explanatory variables covered climate and biophysical variables such as vegetation types, temperature, precipitation, radiation, and satellite-derived fraction of absorbed photosynthetic active radiation. Then the MTE GPP product was generated through applying the trained regression trees for global upscaling using gridded data sets of the same explanatory variables".

3. Figure 1 (on page 23) shows 16 different kinds of vegetation types in the legend but only the major ones are visible in the plot. To make the plot readable, similar vegetation types like MIXSB, MIXSC, MIXSG, SHRUB should be merged since they are anyway not much distinguishable in the plot.

Response: Based on this suggestion, we revised Fig. 1 by merging similar vegetation types like MIXFS, MIXFG, MIXFC, MIXSG, MIXSC, MIXSB, MIXGB, MIXGC. The new Fig. 1 has only 8 different kinds of vegetation types. Please see Page 23.

4. Page 5, Section: 3.1. Since this section starts with the discussion of results presented in Fig. S1 and has an entire paragraph on this figure, the figure should be moved to the main text.

Response: Based on the suggestion, old Fig. S1 has been moved to the main text to be new Fig. 2. Please see Page 24.

5. There is a lot of mismatch between the region references in terms of region names and regions numbers in the Results section. For instance: a. Page 7, line 13, "central China and northern China" should rather be "northern China (R4) and northwestern China (R5)", as per the numbers represented in figure 4. b. Page 7, line 32, "in summer over southeastern China (Fig. 5j)". 5j corresponds to R9 and as per fig. 1,

R9 is southwestern China, not southeastern China. To avoid this confusion in region names and region numbers, I would strongly recommend the authors to double check the text in the sections of Results and Discussions, and to use region numbers along with region names in these sections to that the text explanation can be verified easily with the figures.

Response: Based on the comments, we revised all these sentences by adding the region numbers. Please see Page 7 (Lines 20−21): "The ensemble mean GPP of SG3 over R9 was found to explain the largest fraction (17%) of the IAV for China's GPP, followed by R5 (15%) and R4 (14%)", and Page 8 (Line 6): "almost the same seasonal variations except for a few differences in summer over R9 (Fig. 6j)". The other relative sentences were also revised. Please see the revised manuscript.

6. Page 11, line 12 and Page 1, line 34: A strong concluding statement has been made about how climate is the dominant control factor of annual trends, IAV and seasonality of China's GPP, without much analysis of results in this context in the Results section. Some analysis of trends coming from SG1 case should be included in the results section before making this statement, specifically since the paper has focussed mostly on LULCC and CO2 effects, and there are not many remarks on impact of climate in the paper.

Response: Based on the suggestions, we added some analysis results about the trends from SG1 to the Section 5. Please see Page 11 (Lines 20−22): "In general, climate was the dominant control factor for the trends, interannual variation, and seasonality of China's GPP. When only constrained by climatic driver, mean annual GPP from 1981 to 2010 over China is 6.9±1.7 Pg C yr−1, with a trend of 0.0036 Pg C yr−2".

7. The implications of this study and application of the results are not adequately emphasised. The authors are suggested to add some information on how this work his valuable, specifically considering how understanding of the effects of LULCC and CO2 on GPP can help in comprehensive scenario of things and decision making.

[Figure]

Response: Based on the comments, we added some descriptions about the implications of this study and possible applications of the results to the conclusion section (Section 5). Please see Page 11 (Lines 18−31): "The simulated GPP for China from the 12 MsTMIP models, driven by common climate forcing, LULCC, and CO2 data, was 7.4±1.8 Pg C yr−1, which agreed well with independent MTE data set (7.1 Pg C yr−1). In general, climate was the dominant control factor for the trends, interannual variation, and seasonality of China's GPP. When only constrained by climatic driver, mean annual GPP over China from 1981 to 2010 is 6.9±1.7 Pg C yr−1, with a trend of 0.0036 Pg C yr−2. The overall rise in CO2 enhanced plant photosynthesis and thus increased total China GPP, with increasing annual mean and interannual variability, especially in northeastern and southern China where vegetation is dense. LULCC decreased the IAV of China's total GPP by ∼7%, whereas rising CO2 induced an increase of 8%. Our research examined the joint effects of the three factors and their quantitative contributions to the interannual variations and seasonal cycles of GPP. Given the important role of GPP in regulating terrestrial carbon cycling, this work is expected to help us better understand the interactions of the carbon cycle, climate change, and human activity. Furthermore, it will also be interesting for the policy makers to make public decisions on how to achieve the balance between the optimized economy and minimized carbon loss".

Other issues to be considered:

1. There is no mention of the study period in the abstract so it is not clear for which years are the results mentioned in this section applicable for.

Response: Based on this comment, we added the detailed information about the time periods used in this work. Please see Page 1 (Line 36): "The simulated ensemble mean value of China's GPP between 1981 and 2010".

2. The phrase "independent upscaling GPP estimate" in the abstract does not give any idea of the dataset being talked about and hence should be either modified or

eliminated from this section.

Response: Based on the suggestion, the sentence "which was in close agreement with the independent upscaling GPP estimate (7.1 Pg C yr−1)" has been removed from the abstract. Please see Page 1 (Line 37).

3. The usage of a few words and sentence formation in the text is questionable in some places, for instance: a. Page 2, lines 25 and 27: "60% of the uptake by terrestrial ecosystem was due to raising(?) atmospheric CO2" and "It suggests that the impact of raising(?) CO2 on land carbon sink may be a negative feedback to future climate". b. Page 4, line 14: "The simulated monthly GPP from these 12 models was conducted(?) for the period of 1981−2010." The authors are suggested to re-check these typos and small errors.

Response: Based on this comment, these sentences have been revised in the new manuscript.

(a) Please see Page 2 (Lines 23−24): "Schimel et al. (2014) found that up to 60% of the present-day terrestrial sinks was caused by increasing atmospheric CO2", and Page 2 (Lines 26−27): "It suggests that the CO2 effect on land carbon storage may be a key potential negative feedback to future climate (Schimel et al., 2014)".

(b) Please see Page 4 (Lines 14−15): "The simulated monthly GPP from these 12 models for the period of 1981–2010 was used in this work".

4. Table S1 (mentioned on Page 4, line 10) only has all "O" under columns SG1, SG2 and SG3 for all models, check attached file. I am not sure what purpose the table is serving apart from citing references for each model description. This table can either be improved or deleted.

Response: Based on this comment, we deleted the Table S1 in the supplemental material. Please see Page 1 (Lines 35−57) of the supplemental material.

5. Page 5, line 23, Fig. 1a.(?). This seems to be a typo and Fig. 2a. should be

mentioned here.

Response: Thanks for your suggestion. It has been revised to be "Fig. 3a", since the old Fig. S1 was added to be new Fig. 2. Please see Page 5 (Line 27).

6. Figure 7 has comparison of LUH1 data with CLUD for major vegetation types. Clearly, there is a mismatch in the recent trends of both datasets, more specifically from year 2000 to 2010. This difference is intriguing but since the figure does not represent 100% land cover of China, there is missing information here. For instance, the sum of major vegetation types shown in fig. 7a represents ~60% of area for CLUD for 2010 and ~80% of land cover for LUH1 for 2010. I would suggest this figure to be modified to account for 100% area of China so that the entire land cover distribution and the transitions can be accounted for.

Response: Based on this comment, we added a new type "Other", which includes SNICE (snow and ice), water, and bare soil, to the new Fig. 8. Please see Page 30.

Please also note the supplement to this comment:
https://www.earth-syst-dynam-discuss.net/esd-2019-22/esd-2019-22-AC2-supplement.zip